Generation and characterization of mAb 61H9 against junctional adhesion molecule-a with potent antitumor activity

http://orcid.org/0000-0002-4640-0849 Liu Kang 1 2
Yang Hang 2
Xiong Rong 2
Shen Yunlong 1
http://orcid.org/0000-0002-8068-4742 Song Guiqin 3
Yang Jinliang 1 jinliangyang@scu.edu.cn
Wang Zhenling 1 wangzhenling@scu.edu.cn
1 State Key Laboratory of Biotherapy and Cancer Center, West China Hospital, Sichuan University, and Collaborative Innovation Center of Biotherapy, Sichuan University , Chengdu, Sichuan , China
2 Institute of Tissue Engineering and Stem Cells, Nanchong Central Hospital, The Second Clinical Medical College, North Sichuan Medical College , Nanchong, Sichuan , China
3 School of Basic Medicine and Forensic Medicine, North Sichuan Medical College , Nanchong, Sichuan , China
Date Swapneeta
Electronic publication date: 2024 Mar 14
Publication date: 2024
Volume: 12
Electronic Location ID: e17088
Received 2023 Sep 19; Accepted 2024 Feb 20
Copyright: © 2024 Liu et al.
Copyright year: 2024
Copyright holder: Liu et al.
License: This is an open access article distributed under the terms of the Creative Commons Attribution License, which permits unrestricted use, distribution, reproduction and adaptation in any medium and for any purpose provided that it is properly attributed. For attribution, the original author(s), title, publication source (PeerJ) and either DOI or URL of the article must be cited.
License URL: https://creativecommons.org/licenses/by/4.0/

Keywords: Esophageal cancer, Junctional adhesion molecule-A, Anti-JAM-A, Monoclonal antibody, Hybridoma

Funding: National Natural Science Foundation 82203851 Sichuan Science and Technology Program 2023NSFSC0731, 2023YFSY0045 Nanchong Science and Technology Program 22SXQT0336,20SXQT0328 This work was supported by the National Natural Science Foundation of China (82203851), the Sichuan Science and Technology Program (2023NSFSC0731, 2023YFSY0045), and the Nanchong Science and Technology Program (22SXQT0336, 20SXQT0328). The funders had no role in study design, data collection and analysis, decision to publish, or preparation of the manuscript.

==============================
Junctional adhesion molecule-A (JAM-A) is an adhesion molecule that exists on the surface of certain types of cells, including white blood cells, endothelial cells, and dendritic cells. In this study, the cDNA sequences of JAM-A-Fc were chemically synthesized with optimization for mammalian expression. Afterward, we analyzed JAM-A protein expression through transient transfection in HEK293 cell lines. Mice were immunized with JAM-A-Fc protein, and hybridoma was prepared by fusing myeloma cells and mouse spleen cells. Antibodies were purified from the hybridoma supernatant and four monoclonal strains were obtained and numbered 61H9, 70E5, 71A8, and 74H3 via enzyme-linked immunosorbent assay screening. Immunofluorescence staining assay showed 61H9 was the most suitable cell line for mAb production due to its fluorescence signal being the strongest. Flow cytometric analysis proved that 61H9 possessed high affinity. Moreover, antagonism of JAM-A mAb could attenuate the proliferative, migrative, and invasive abilities of ESCC cells and significantly inhibit tumor growth in mice. By examining hematoxylin-eosin staining mice tumor tissues, we found inflammatory cells infiltrated lightly in the anti-JAM-A group. The expression of BCL-2 and IκBα in the anti-JAM-A group were decreased in mice tumor tissues compared to the control group. Ultimately, a method for preparing high-yield JAM-A-Fc protein was created and a high affinity mAb against JAM-A with an antitumor effect was prepared.

Introduction

Esophageal cancer (EC) is a common and fatal cancer with obvious geographical variations in the incidence, mortality, and histopathology. It is a severe public health burden that impairs quality of life, especially in Eastern/Southern African and East Asian countries (Collaborators, 2020; Peery et al., 2022). Esophageal adenocarcinoma (EAC) and esophageal squamous cell carcinoma (ESCC) are two histological subtypes that account for the majority of ES cases (Rogers et al., 2022; Salem et al., 2018). Worldwide, more than 85% of EC cases are ESCC (Arnold et al., 2017). In the past few decades, some progress has been made in the treatment of this cancer. At present, the first-line treatment is mainly doublet platinum-based chemotherapy, with trastuzumab in human epidermal growth factor receptor two (HER2)-positive ES (Smyth et al., 2021), but its efficacy is very limited, and the most common adverse reactions include hematological toxicity (myelosuppression), gastrointestinal toxicity, alopecia, and sensory neuropathy (Kojima et al., 2020). Many metastatic ES patients receive only first-line therapy, meaning that they do not have the chance to benefit from novel therapies (Janjigian et al., 2020). For advanced ESCC, no targeted drugs were approved until recently, when Nivolumab was approved as a second-line chemotherapy treatment after first-line drug resistance (Kang et al., 2017). Therefore, efforts should be made to develop a new therapeutic drug.

Junctional adhesion molecule-A (JAM-A), an immunoglobulin superfamily (IgSF) protein located at epithelial and endothelial tight junctions, plays an important role in cell permeability, migration, and polarity (Ebnet et al., 2001; Martìn-Padura et al., 1998). In pathophysiology, studies have reported JAM-A to be overexpressed in different cancer types including breast, gastric, lung, nasopharyngeal, and brain cancer, as well as hematological malignancies (Brennan et al., 2013; Leech et al., 2018; McSherry et al., 2009; Murakami et al., 2011; Tian et al., 2015; Zhang et al., 2013). Overactivation of JAM-A is caused by upregulation or abnormal dimerization, which puts the receptor in a constitutive signaling state, or excessive release of JAM-A ligands into the microenvironment by normal and tumor cells (Rajkumar et al., 2014). The membrane-bound JAM-A can form homologous interactions with afadin (AFDN) and lymphocyte function-associated antigen 1 (LFA-1), as well as heterophilic interactions. Calcium/calmodulin dependent serine protein kinase (CASK) have high receptor/ligand binding affinity (Rajkumar et al., 2011). These responses trigger the downstream signaling pathway of JAM-A, which is involved in regulating the survival, growth, angiogenesis, and propagation of tumor cells. However, the role of JAM-A is unclear in esophageal cancers, and there has been debate on the aberrant expression of JAM-A in this context (Richards et al., 2021). It is unclear whether JAM-A reflects inter- or intra-tumor heterogeneity (Kakogiannos et al., 2020). Xiong et al. (2023) demonstrated that the expression of JAM-A was high in ESCC, and the cell cycle arrest at G1 was induced and proliferation, invasion, and migration were suppressed by JAM-A knockdown.

JAM-A is the key point in endothelial cell pathology and physiology, and its expression has been proved necessary for self-renewal and tumor growth (Schellerer et al., 2007; Solimando et al., 2018). Therefore JAM-A dysregulation within tumors appear to be tissue and organ-dependent, which has important implications for the use of JAM-A as a therapeutic target (Harryvan et al., 2022). However, there is currently no anti-JAM-A monoclonal antibody used to treat ESCC.

In this study, JAM-A was expressed by eukaryotic expression systems and all JAM-A-Fc proteins had great biological activity. Additionally, as an antigen protein, JAM-A-Fc was used to screen hybridomas that could bind to JAM-A directly and developed monoclonal antibodies for the treatment of ESCC. We found a high-affinity monoclonal antibody 61H9G4 against JAM-A through western blot (WB) and immunofluorescence (IF). In addition, compared with commercial antibody FITC-anti JAM-A, 61H9G4 couldtarget JAM-A protein at different sites. We also demonstrated that 61H9G4 had sufficient function to reduce cancer cell activity in vivo and inhibit tumor growth in vivo, which indicated the therapeutic potential of this construct.

Materials and Methods

Materials and animal experiments

The JAM-A-Fc cDNA and expression vector pTT5 were supplied by Corebiolab (Wuhan, China), as well as all molecular cloning reagents, including DNA polymerase, DNA ligation, and EcoRI/XhoI. Human ESCC cells (KYSE30, KYSE410) were purchased from Procell Life Science & Technology (Hyderabad, India). Mouse myeloma cells were obtained from Corebiolab.

The nude male mice and BALB/C mice were acquired from Beijing Huafukang Biotechnology Co., Ltd, Beijing, China. The mice were kept in the room with a temperature of 22 ± 2 °C, under light and dark for 12 h. Three to five mice were housed in each cage with free access to food and water. This project used 17 animals for modeling by subcutaneous injection. In the first animal experiment, we used five male BALB/C mice to explore monoclonal preparation. In the following experiment, 12 BALB/c nude male mice were disrupted before grouping. This represented a reasonable method for generating randomized sequences, by randomly dividing the animals into a control group and experimental group, with six mice in each group. The mice used in the experiment were all male, 1–2 months old, and weighed about 20 g. For different experiments, samples were processed in different ways, and all mice were finally sacrificed by intravenous injection of anesthetic drugs. The measurements included mouse weight, serum titer, and tumor size. Euthanasia was carried out when animals lost 25% of their weight, lost appetite completely for 24 h, or could not stand or could only stand with extreme reluctance due to physical weakness for 24 h. Any of the above conditions were the end point of the observation of the project. Animal experiments were conducted in the North Sichuan Medical College (NSMC) animal house without specific pathogens and in accordance with the requirements of the Animal Care and Use Committee of NSMC. The IACUC approval number was ECNSMC202144.

JAM-A-Fc recombinant protein expression and purification

In order to acquire JAM-A-Fc protein, the Fc tag was fused at the C-terminus of the sequence and the sequences were chemically synthesized with optimization for mammalian expression. The cDNA was cloned in expression vector pTT5-Fc and endotoxin-free DNA preparation was done for the pTT5 expression construction obtained. By using a proprietary method, the plasmid was then transiently transfected into HEK293 cells. Culture medium and cells were collected until viability dropped under 50%. Purification was conducted by affinity vs. Protein A resin, equilibration, and washing with PBS buffer (pH 7.5), then elution with 20 mM citric acid buffer (pH 2.7). Analysis was carried out SDS-PAGE of fractions of interest, qualitative and quantitative.

Preparation of mouse monoclonal antibody

Balb/c mice aged 6–8 weeks were purchased for antigen immunization. We mixed 100 μL JAM-A-Fc protein with equal volume Freund’s complete adjuvant for primary immunization on mice via back and intraperitoneal injection. After an interval of 2 weeks, the same treatment was carried out for immunization. One week after the third and fourth immunization, the serum titers of mice were detected by ELISA. If a titer did not reach expectations, the fifth booster immunization was carried out. Seven days after the fourth injection, spleen cells were obtained and fused with myeloma cells. We used 1 mL of 50% polyethylene glycol (PEG) to evenly mix myeloma cells (>107) and immune spleen cell suspension (1 × 108), and then the mix was transferred to a 96-well cell culture plate and cultured in a 37 °C constant temperature incubator with 5% CO2. After fusion, we observed the growth of hybridoma cells, and aspirated the supernatant for ELISA detection when the cells grew to 1/4–1/3 of the hole bottom. P/N ≥ 2.5 was considered as the positive judgment standard, and the serum of the non-immunized mice was the negative control, and the antibody dilution was the blank control. The positive cells were expanded and cloned. A limited dilution method was used for cloning culture. We selected the positive wells with few clones and high OD450 nm values and cloned them again. After one to two cloning operations, when the positive rate of all cloned cell pores reached 100%, the hybridoma cell line secreting specific monoclonal antibody could be determined, expanded, and timely frozen.

Enzyme-linked immunosorbent assay (ELISA)

JAM-A-Fc (1 μg/mL, 100 μL/well) was placed in coated microplates (Corning, Corning, NY, USA), incubated in a 37 °C incubator for 60 min, then removed and washed with PBST three times. Diluted serum was added into each plate and incubated in a 37 °C incubator for 60 min. We added 100 μL secondary antibody goat anti-mouse IgG (Jackson, West Grove, PA, USA), incubated in 37 °C for 1 h, and then washed with PBST (300 μL/well). Finally, TMB was added into the microplates. We put the microplates into the incubator for 5–10 min, then added 2 M HCl into the substrate and later read absorbance at 450 nm.

Immunofluorescence staining

The counted cells were seeded in a six-well culture plate with coverslip so that the coverslip was completely immersed in the medium. The plates were incubated in a 5% CO2 incubator at 37 °C for 2–3 days. When adherent cells covered 2/3 of the plate bottom, we took out the plates and soaked the slides with PBS three times for 3 min. After fixation with 4% paraformaldehyde for 15 min, normal goat serum was added dropwise to the slides and blocked for 30 min at room temperature; the blocking solution was blotted off using absorbent paper, and adequate primary antibodies were dropped to each slide and put into a wet box and incubated overnight at 4 °C. Then slides were immersed and washed three times for 3 min each with PBST, and the fluorescent secondary antibody was added dropwise after blotting the excess liquid with absorbent paper. After incubating at 20–37 °C for 1 h in a wet box, slides were immersed and washed with PBST three times for 3 min each time. DAPI was then added dropwise and incubated for 5 min in the dark, followed by a blocking solution containing an anti-fluorescent quencher four times for 5 min each to block the sections. Fluorescence microscope was used for observation and acquisition.

To optimally visualize inflammatory cells, our evaluation was based on hematoxylin-eosin staining. The samples were fixed by 95% ethanol for 20 min and washed twice with PBS for 1 min each. Nuclei were stained with hematoxylin stain for 2–3 min and washed with PBS. To stain the cytoplasm, eosin was stained for 1 min and washed with PBS. After blowing or naturally drying the stem cells to climb the pieces, the neutral gums covered the pieces.

Flow cytometric analysis of antigen sites

KYSE30 cells were cultured with RPMI-1640 containing 10% fetal bovine serum, penicillin (100 μ/mL), and streptomycin (100 μ/mL) at 37 °C in saturated humid conditions with 5% CO2. All cells used in the experiment were in the log growth phase. For flow cytometry, normal cultured KYSE30 cells were collected and incubated with corresponding antibodies for 30 min in the dark at 4 °C. After incubation, cells were washed and collected via centrifugation at 1,000 rpm for 5 min. This process was repeated twice. After the last centrifugation, we removed supernatant and added 100 μL PBS to resuspend cells. Finally, the samples were analyzed using NovoExpress software to identify the antigen sites of 61H9G4 mAb.

Cell proliferation assay

KYSE30/KYSE410 cells were placed into 96-well plates and observed daily, and hybridoma supernatant was added to the cell culture plate after gradient dilution (no antibody was added to the control group). We added 10 μL of CCK-8 solution (Dojindo Laboratories, Shanghai, China) into each well and incubated at 37 °C for 3 h. After 24, 48, 72, 96, and 120 h, cell proliferation was assessed by CCK8 assay in each respective group.

Cell apoptosis analysis with flow cytometry

According to FITC and PI fluorescence values, two fluorescence parameters were determined to delimit the cross gate. In the experiment, cells were divided into three subpopulations: living cells that were double negative (Annexin v-FITC-/PI-), early apoptotic cells that were annexin V-FITC single positive (Annexin v-FITC+/PI-), and late apoptotic cells that were double positive for annexin V-FITC and PI (Annexin v-FITC+/PI+). The apoptosis rate of our experimental results was in the upper right quadrant (late apoptosis) and the lower right quadrant (early apoptosis). We added 61H9G4 mAb or IgG to KYSE30/KYSE410 cells, and the next steps were implemented according to the instructions of the Annevix FITC Kit (Vazyme, Nanjing, China). A NovoCyte flow cytometer and NovoExpress software were used for sample analysis and subsequent analysis, separately.

Cell cycle analysis with flow cytometry

Different antibodies were added to KYSE30/410 cells for 72 h. Cells were collected and washed once with PBS. Cell suspension was added into pre-cooled ethanol at −20 °C, and then was fixed at 4 °C overnight, 1,000 rpm * 5 min. We removed the ethanol, added 200 μL DNA dye to each tube, kept them away from light at room temperature for 15 min, and used the NovoCyte flow cytometer for detection.

Scratch-migration assay

Two × 105 cells were seeded into six-well plates and grown to confluency. After cells were plated into the plate bottom, a cell scratch was made with a 10 μL pipette tip perpendicular to the well plate, ensuring that individual scratch widths were consistent. The cell culture medium was aspirated, well plates were rinsed with PBS three times, cell debris generated by the scratch was washed away, and plates were observed and imaged after 0, 6, 24 and 48 h under microscope (IX71; Olympus, Shinjuku City, Tokyo, Japan).

In vivo experiments

BALB/c nude mice (male, 5 weeks) were randomly divided into anti-JAM-A and control groups (n = 6 each group) subcutaneously for a total of 3 × 106 KYSE30 cells to initiate tumors. Thirteen days after cell inoculation, control and treated mice were intratumorally and peripherally injected with PBS and 61H9G4, respectively, every 2 days for a total of six injections. Tumor volume size was measured every 3 days. The mice were sacrificed after 39 days, and the tumor tissues were collected.

Statistical analyses

GraphPad prism 9.0 was used for statistical analysis and plotting, the measurement data were expressed by mean ± S.D. T-test or one-way ANOVA were used to compare two or more groups. The difference was statistically significant with P<0.05 (*P < 0.05, **P < 0.01, ***P < 0.001). NS stands for no statistical significance. The assumptions for the t-test included normal distribution, homogeneity of variance, and independence. The Shapiro Wilk test was used to check for normality, and the Levene test was used to check for homogeneity of variance.

Results

Expression and purification of JAM-A-Fc recombinant protein

The antigen sequences were chemically synthesized (Corebiolab, Wuhan, China) and the structure of JAM-A gene in a linear model was presented (Fig. 1A). The constructed JAM-A expression plasmids pTT5-JAM-A-Fc were transfected into the mammalian expression cell HEK293 with transfection reagent, and on the 10th day after transfection, we collected the cell supernatant for purification. The cell supernatant was purified by affinity vs. protein A resin, the purity of JAM-A-Fc protein was about 90% and its molecular weight was about 56.4 kDa with a yield of 3.93 mg (Fig. 1B).

Figure 1 Expression and purification of recombinant JAM-A-Fc.

(A) Schematic drawing of construct containing the sequence of JAM-A followed by a C-terminal human IgG1 Fc tag. (B) SDS-PAGE analysis of purified JAM-A-Fc proteins. 2 μg of purified production were loaded onto 10% SDS gels.

Preparation and identification of anti-JAM-A monoclonal antibodies

Eight fusion cells were screened by limited dilution assay. Clones with OD450 > 0.5 were selected and retested in new wells. Fc labeled antigen was added as negative screen control to eliminate Fc labeled antibody. Finally, four monoclonal strains with high JAM-A-Fc OD450 value and low Fc OD450 value were obtained and named 61H9, 70E5, 71A8, and 74H3, respectively. Immunofluorescence staining assay showed 61H9 was the most suitable cell line for mAb production as its fluorescence signal was the strongest (Figs. 2A, 2B).

Figure 2 Endogenous detection of monoclonal cell lines by immunofluorescence staining.

(A) Images for the expression of JAM-A were captured by fluorescence microscope. DAPI staining solution is often used to stain nuclei, which can stain blue. Scale bar, 20 μm. (B) The relative fluorescence intensity was examined by immunofluorescence staining ( Unpaired t-test ). **P < 0.01,***P < 0.001

The obtained anti-human JAM-A mouse mAb hybridoma variable region sequence (61H9) was subjected to antibody full-length recombinant expression following splicing to the constant region sequence. The above two pieces of sequences were respectively constructed into the expression vector pTT5 and transfected into CHO cells for antibody expression, and supernatant and cells were harvested for protein A antibody affinity purification. After using the reductive loading buffer to open the disulfide bond, there were approximately 55 (heavy chain) and 25 kD (light chain) bands, indicating that the antibody is an IgG type antibody (Fig. S1A), and purified antibodies were also present under non-reductive conditions (Fig. S1B). Finally, 23 mg of anti-JAM-A (61H9G4) was obtained for subsequent antibody function experiments.

Evaluation of antibody binding site of 61H9G4

To compare the difference between 61H9G4 and the commercial antibody AB275688 that binds to JAM-A, we carried out flow cytometric analysis. By adding fluorescein, the %parent of 61H9G4 and AB275688 binding to JAM-A was ascertained. The experimental results indicated that the % parent of the control group, AB275688, and 61H9G4 were 0.09%, 99.98%, and 99.33%, respectively (Figs. 3A–3C). Competitive inhibition binding assays was performed by first incubating homemade antibodies and then incubating commercial antibodies with a %parent of 30.89% (Fig. 3D), which showed that the recognition sites of JAM-A antigen by self-made antibody and commercialized antibody had some differences. Additionally, the binding activity of 61H9G4 to the JAM-A site was similar to that of AB275688, both of them have significantly higher %parent than control group (Fig. 3E).

Figure 3 Identification of antigen sites of JAM-A monoclonal antibody by flow cytometry.

(A) IgG, negative control group. (B) Commercial antibody FITC-anti-JAM-A (AB275688) was incubated separately, positive group 1. (C) Incubated self-made monoclonal antibody 61H9G4 separately, positive group 2. (D) Self-made antibody was first incubated and then commercial antibody was incubated, competitive inhibition binding test group. (E) Quantification of binding antigen sites of 61H9G4 and AB275688 with JAM-A (Unpaired t-test). ****P < 0.0001

61H9G4 inhibited proliferation and migration, and promoted apoptosis of ESCC cells

In the CCK-8 assays, cell viability decreased 24, 48, 72, 96, and 120 h after 61H9G4 treatment compared with the control groups (0 μg/mL). Specifically, when the concentration of 61H9G4 was 40 μg/mL, the cell viability decreased fastest. This indicated that 61H9G4 exerted a proliferation-suppressive effect on both KYSE30 and KYSE410 cells (Fig. 4A). To explore whether 61H9G4 affected cell apoptosis, we used flow cytometry to compare the capacity of 61H9G4 vs. IgG to induce apoptosis. From the results, we found that 61H9G4 significantly promoted cell apoptosis compared with IgG (P < 0.001) (Fig. 4B). Furthermore, we explored the influence of 61H9G4 on cell cycles using flow cytometry. Notably, 61H9G4 arrested cells in the G0/G1 phase, which may promote cell apoptosis (P < 0.01 or P < 0.001) (Fig. 4C). Meanwhile, we analyzed scratch wound healing, and calculated the relative migration distance at 0, 6, 24, and 48 h. As shown in Fig. 5A, cell migration ability significantly decreased in the 61H9G4 group compared to the IgG group (P < 0.05). The cell migration ability in the 61H9G4 group was significantly weaker than the control group at 24 h (P < 0.01). This showed that 61H9G4 could suppress migration of esophageal squamous cell carcinoma. In the transwell assay, after 24 h of cell culture, the cell counts of the 61H9G4 group was significantly decreased compared with the IgG group (P < 0.01) (Fig. 5B).

Figure 4 JAM-A mAb inhibited proliferation, promoted apoptosis of ESCC cells.

(A) Proliferation ability of JAM-A mAb in KYSE30 and KYSE410 was assessed by CCK-8 assay. (B) The cell apoptosis degree of JAM-A mAb in KYSE30 and KYSE410 was analyzed by flow cytometry ( Two-tailed t-test ). (C) The cell cycle of KYSE30 and KYSE410 with or without JAM-A mAb (Two-tailed t-test). **p < 0.01, ***p < 0.001.

Figure 5 JAM-A mAb inhibited migration of ESCC cells.

(A) Scratch-migration assay at 0, 6, 24, and 48 h after adding JAM-A mAb or IgG in ESCC cells. Scale bar, 100 μm (two-tailed t-test). (B) Transwell invasion assays were carried out in KYSE30 and KYSE410 with JAM-A mAb or IgG (two-tailed t-test). Scale bar, 50 μm. *P < 0.05, **P < 0.01, ***P < 0.001

Effects of 61H9G4 on cell proliferation and invasion related pathways

We used WBto reveal the possible proliferation- and invasion- related signaling pathways affected by anti-JAM-A (Figs. 6A, 6B). In proliferation-related analysis, we found that 61H9G4 can significantly inhibit the expression levels of CyclinD1 and BCL2 in cells (Figs. 6C, 6D). CyclinD1 is a key protein in regulating the G1 phase of the cell cycle, and overexpression of CyclinD1 is closely related to the occurrence and development of malignant tumors. BCL2 can inhibit apoptosis occurrence. In addition to that, 61H9G4 could significantly increase the expression levels of p53 and caspase-3 in cells, and p53 acts as a typical tumor suppressor gene and can inhibit cell proliferation and migration. Additionally, cell apoptosis is accompanied by caspase-3 activation.

Figure 6 Effects of monoclonal antibodies on cell proliferation and invasion related pathways.

(A) Western blot analysis for proliferation and invasion-related protein in KYSE30. (B) Western blot analysis for proliferation and invasion-related protein in KYSE410. β-Tubulin was used as loading control. (C) Quantification of relative protein level KYSE30. (D) Quantification of relative protein level KYSE410. (Two-tailed t-test), ***P < 0.001

The NF-κB pathway is a classical signal pathway that promotes tumor genesis and metastasis. By detecting the phosphorylation level of IκBα and P65, two typical indicators of this pathway, we found that 61H9G4 can significantly inhibit phosphorylation of IκBα and P65 proteins, which may inhibit NF-κB pathway activation and thereby regulate cell related phenotypic changes such as proliferation and migration. The above results indicated that 61H9G4 was closely related to the regulatory pathways that inhibit the proliferation and invasion of ESCC.

61H9G4 effectively inhibited KYSE30 cell growth and decreased BCL-2 and IκBα expression

We further investigated the antitumor efficacy of 61H9G4 in vivo. After 39 days implantation, the tumors were dissected from the BALB/c nude mice to measure the volume (Fig. 7A). As shown in Fig. 7B, from the 30th day of cell inoculation, the tumor growth of the treatment group began to slow down, and the tumor volume was significantly inhibited from the 33rd day. On the 39th day, the inhibition rate of tumor growth in the 61H9G4 treatment group was about 50% compared with the control group. This suggested that 61H9G4 treatment effectively inhibited tumor growth in mice.

Figure 7 JAM-A antibody function experiment of Balb mice tumor-bearing model and different expression in immunohistochemistry and hematoxylin-eosin staining.

(A) Comparison of tumor volume in the control group and JAM-A mAb group. (B) Tumor growth curves, 10 days after cell inoculation, the tumor volume was measured and tumor size was recorded every 3 days (paired t-test). (C) The tumor cell morphology of nude mice tumor tissues in the control and JAM-A mAb group were detected by HE staining. (D) Immunohistochemical staining for the expression of BCL-2 and IκBα in nude mice. (E) Quantification of overall BCL-2 and IκBα staining intensity in control group and anti-JAM-A group. (Two-tailed t-test), *P < 0.05, **P < 0.01, ***P < 0.001

Additionally, compared with the control group, HE staining showed that cancer cells in the 61H9G4 group had small nuclei, light staining, and reduced mitotic stages (Fig. 7C). Consistent with the in vitro experimental results, immunohistochemistry analysis showed that anti-JAM-A group had significantly lower expression of BCL-2 and IκBα than control group in nude mice tumor tissue (Figs. 7D, 7E), was related to the tumor volume of nude mice.

Discussion

EC places a heavy burden on people’s life and mental state along with enormous public health and economic burden (Bray et al., 2018; Ishigaki et al., 2016). For resectable EC, the main treatment is surgery with/without neoadjuvant, adjuvant chemotherapy, or chemoradiotherapy (Han, Wang & Liu, 2021; Jing et al., 2019; Zhang et al., 2022). For inoperable patients, targeted therapy, radiotherapy, chemotherapy, and immunotherapy may be treatment options (Nakatani et al., 2020; Simoni et al., 2020). The pathogenesis of EC has not been elucidated, and multiple factors may be involved in its pathogenesis and mortality (Li et al., 2023). In recent years, a large number of epidemiological studies have revealed that the occurrence of EC is the result of a long-term interaction between genetic and environmental factors. Because most EC patients are in the intermediate to advanced stages of the disease, treatment options are limited (Thrift, 2021). Therefore, research into new treatment modalities is highly necessary. Antibody drug-targeted therapy has been widely used in tumor-targeted therapy, organ transplantation, and autoimmune diseases due to its strong targeting ability and few side effects (Kumar & Mahal, 2012).

JAM-A was initially characterized as a cell junction protein that plays an important role in preserving epithelial cell organization and tissue integrity (Naik et al., 2001). Additionally, JAM-A exerts these functions through a homophilic combination on adjacent cells, strengthening integrin effect on the same cell, and interacting with the integrins on adjacent cells (Steinbacher, Kummer & Ebnet, 2018).

Currently, cancer research has demonstrated that JAM-A regulates the tumorigenesis and anti-tumor processes in a specific manner and can be used as a biomarker for some types of cancer (Zhao et al., 2017). Previous studies have shown that the down-regulation of JAM-A expression leads to the loss of cell adhesion and epithelial barrier function, thereby increasing cell permeability and causing tumor cell invasion and metastasis (Danthi et al., 2006; Ostermann et al., 2005). In recent years, many studies have shown that the up-regulated expression of JAM-A drives tumorigenesis and promotes metastasis and drug resistance by activating cell adhesion independent signaling pathways (Leech et al., 2015). Our previous studies showed that the expression of JAM-A was up-regulated in ESCC, and the reduction of JAM-A expression inhibited the growth and metastasis of tumors. At the same time, in this study, the prepared JAM-A antibody also had the same anti-tumor effect.

In this study, we constructed the pTT5 expression vector and purified 3.93 mg recombinant JAM-A-Fc protein from a HEK293 cell culture. As an antigen, JAM-A-Fc provided the immune basis for hybridoma-immunized mice, and JAM-A-Fc was sufficient for the next stage of ELISA screening. The affinity of 61H9G4 with JAM-A was analyzed by flow cytometric analysis and the results are shown in Fig. 3. The %parent of the control, 61H9G4, and commercial antibody AB275688 groups were 0.09%, 99.33%, and 99.98%, respectively. These results demonstrated that antibody 61H9G4 could stably identify and bind JAM-A. In addition, 61H9G4 and AB275688 incubated with a %parent of 30.89%, and the results showed that the binding epitopes of 61H9G4 and AB275688 were different.

As far we know, this is the first study on tumor target therapy that looked at JAM-A as a biomarker in ESCC cells, and we found that 61H9G4 could decrease the proliferative, migrative, and invasive abilities of ESCC cells. Similarly, it could significantly reduce the number of mitotic cells in KYSE30 and KYSE410. Furthermore, the antitumor effect of 61H9G4 on a mouse transplanted tumor model was evaluated. At the animal level, it was able to significantly inhibit tumor growth, and have good therapeutic effects. NF-κB is a transcription factor that regulates the expression of a large number of genes that are essential for regulating tumorigenesis, viral replication, inflammation, and autoimmune diseases (Suhail et al., 2021; Sun, 2017). To further dissect the molecular mechanism of JAM-A mAb, we found that 61H9G4 significantly inhibited IκBα and p65 protein phosphorylation, so it was possible that JAM-A mAb inhibited the NF-κB signaling pathway involved in the development of ESCC. However, whether 61H9G4 interacts with NF-κB should be further analyzed. At present, there have been few reports in the literature that the monoclonal antibody targeting JAM-A can resist tumors. Through in vivo and in vitro experiments, it was confirmed for the first time that 61H9G4 can significantly inhibit the growth and metastasis of ESCC, which has a profound historical significance in clinical application. Previous research has demonstrated that JAM-A antagonistic peptide 4D (P4D) can reduce soluble JAM-A level by blocking TEM and adhesion in breast cancer cells (Bednarek et al., 2020). However, there is a lack of in vivo and clinical studies to test the effectiveness of P4D as an anti-tumor drug. In future studies, we will look for the downstream target genes regulated by JAM-A, further study the mechanism of action between them, and explore the cytotoxic effect and immune effector function of JAM-A mAb on tumor cells. Some studies have shown that blocking JAM-A on multiple myeloma cells can restore angiogenic homeostasis and inhibit tumors, that JAM-A mAb has high specificity and target binding affinity (Solimando et al., 2021), and is expected to develop into an ADC (Antibody Drug Conjugate) for multiple myeloma.

Conclusion

In conclusion, we found that JAM-A can be regarded as a potential target in ESCC and designed a method for producing a high affinity mAb 61H9G4 against JAM-A, which can inhibit the growth of ESCC in vitro and vivo. Our study provided some new evidence that 61H9G4 might be a potential candidate for the treatment of ESCC.

Supplemental Information

Supplemental Information 1 Checklist.

Supplemental Information 2 Expression and Purification of Recombinant JAM-A-His.B SDS-Page of JAM-A-His protein.

Supplemental Information 3 Western blot analysis for proliferation, migration and apoptosis related protein expression with F11R mAb or IgG (KYSE30).

Supplemental Information 4 Western blot analysis for proliferation, migration and apoptosis related protein expression with F11R mAb or IgG (KYSE410).

Supplemental Information 5 Preparation of monoclonal antibodies against JAM-A.

Supplemental Information 6 Endogenous detection of monoclonal cells by Western blot.

Supplemental Information 7 Characterization of the purified JAM-A mouse monoclonal antibody by SDS Page.

(A) Purified antibody was reductive with β-mercaptoethanol buffer. Two bands are shown around 55kD (heavy chain) and 25kD (light chain). (B) Purified antibody under non-reductive conditions . 2 μg of purified JAM-A mouse monoclonal antibody were loaded onto 10% SDS-PAGE.

Supplemental Information 8 Endogenous detection of monoclonal cells by Western blot.

The prepared antibody to detect the endogenous protein of cells, and a positive antigen protein as the control.

Additional Information and Declarations

Competing Interests

Author Contributions

Animal Ethics

Data Availability

The authors declare that they have no competing interests.

Kang Liu conceived and designed the experiments, performed the experiments, prepared figures and/or tables, and approved the final draft.

Hang Yang conceived and designed the experiments, performed the experiments, analyzed the data, prepared figures and/or tables, and approved the final draft.

Rong Xiong conceived and designed the experiments, performed the experiments, prepared figures and/or tables, and approved the final draft.

Yunlong Shen conceived and designed the experiments, performed the experiments, analyzed the data, prepared figures and/or tables, and approved the final draft.

Guiqin Song conceived and designed the experiments, prepared figures and/or tables, and approved the final draft.

Jinliang Yang conceived and designed the experiments, authored or reviewed drafts of the article, and approved the final draft.

Zhenling Wang conceived and designed the experiments, authored or reviewed drafts of the article, and approved the final draft.

The following information was supplied relating to ethical approvals (i.e., approving body and any reference numbers):

NSMC Ethical Approval for Research Involving Animals [2021] 44.

The following information was supplied regarding data availability:

The data is available at Figshare: Liu, Kang (2023). Supplemental File_2.7.rar. figshare. Figure. https://doi.org/10.6084/m9.figshare.24161931.v2

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
