# Peer review of "Generation and characterization of mAb 61H9 against junctional adhesion molecule-a with potent antitumor activity"

_PeerJ, doi:10.7717/peerj.17088_

## Round 0.1 · original submission · Major Revisions

Please address all comments from the reviewers.

**Language Note:** The review process has identified that the English language must be improved. PeerJ can provide language editing services - please contact us at copyediting@peerj.com for pricing (be sure to provide your manuscript number and title). Alternatively, you should make your own arrangements to improve the language quality and provide details in your response letter. – PeerJ Staff

Reviewer 1 ·

Basic reporting

1. Modify the title to better reflect the novelty of 61H9 in comparison to other JAM-A targeting antibodies.
2. Refine the conclusion to provide a concise summary of the key findings, avoiding the repetition of background information. Consider including additional supporting data as supplementary information to enhance the completeness of the study.
3. Enhance the introduction with essential background information on JAM-A's role in cancer and the model systems used for study. Include relevant references to support the discussion.

4. Provide comprehensive information in figure legends, including details about statistical analysis tests and the experiments the results are derived from.
5. For Figure 1A, consider presenting the structure of the JAM-A-Fc gene in a linear model figure rather than the structure map of the antigen expression vector plasmid.
6. In Figure 1B, use arrow marks to indicate the JAM-A-Fc protein band and clarify the meaning of "2ug" in the figure or legend.
7. Create a quantitative bar graph in Figure 2 to demonstrate the immunofluorescent signal strength of JAM-A panel images more clearly.
8. Similar to Figure 2B, provide additional information about the details of Figure S1.
9. Add an experimental design label to Figure 3 to aid in understanding the differences among the four figures.
10. Improve the resolution of Figure 4 for better clarity.
11. For Figure 4B, provide the gate of the quantification on the flow cytometry diagrams to enhance the understanding of the statistical analysis of the apoptosis rate.
12. Consider presenting a set of simple bar graphs in Figure 5 to show the quantification of bands more straightforwardly.

13. Avoid redundant expressions, such as "after then" (Line 56).

Experimental design

1. Emphasize the novelty and advantages of 61H9 compared to other JAM-A-targeting agents by highlighting the lack of comparison with existing literature.
2. Clarify the rationale behind the choice of 61H9 and provide a more comprehensive explanation, considering the limited number of clones screened and the absence of quantitative comparison of their binding affinities.
3. Provide a plausible explanation for the right migration peak in the anti-JAM-A panel in KYSE30 cells in figure 4C
4. Include a positive control for JAM-A-Fc in the blots (Figure 1B, S1).
5. Improve the logical flow and transition in certain sections of the results, such as Figure 1B and S1, to ensure a smooth narrative for readers.

Validity of the findings

1. Figure 4C: The observed increase in cells in the G0/G1 phase due to the anti-JAM-A antibody may need further validation, considering the subtle difference despite statistical significance and the similarities in the overall histogram patterns.
2. There might be a need for further clarification regarding the statement in the text suggesting that "The experimental results indicated that %parent of control group, AB275688, 61H9G4 were 0.09%, 99.98%, and 99.33%, respectively (Fig. 3A-C), which indicated 61H9G4 was superior to AB275688 binding to JAM-A site." This statement might require additional context, considering the subtle difference between 99.98% and 99.33%.
3. The discussion section could benefit from additional insights regarding future research directions and potential avenues for further investigation.

·

Basic reporting

In the present work the authors have generated and characterized a high-affinity monoclonal antibody against Junctional adhesion molecule-A. To my best knowledge, the data is novel, interesting and potentially relevant.
1. There are a fair amount of grammatical errors that need to be corrected. The paper should be revised in more scientific language
2. Please include references for lines 334-336, 341

Experimental design

Research question well defined, relevant & meaningful
Methods described with sufficient detail & information to replicate.

Validity of the findings

3. Fig 3, 5 and 6D, are not illustrative. Please include charts illustrating the differences between groups. P-values also should be included in charts.
4) The authors are invited to clarify the sentence, “downregulation of JAM-A expression causes loss of epithelial barrier and cell adhesion function, which increases cell permeability, leading to tumor cell metastasis and invasion of multiple cancer types”. How could this antibody have beneficial effects while it can cause an increase in tumor metastasis by targeting JAM-A?

---

## Round 0.2 · Minor Revisions

Please address the reviewer's comment.

Reviewer 1 ·

Basic reporting

The authors addressed all my comments.

Please ensure the clarity of Figure 4 in the published version. If necessary, consider dividing the figure into two smaller sets for better demonstration.

Experimental design

The authors addressed all my comments.

Validity of the findings

The authors addressed all my comments.

---

## Round 0.3 · accepted · Accept

Your manuscript has been Accepted for publication. Congratulations!